# What’s New in the Treatment of Non-Alcoholic Fatty Liver Disease (NAFLD)

**DOI:** 10.3390/jcm12051852

**Published:** 2023-02-26

**Authors:** Marcin Kosmalski, Rafał Frankowski, Sylwia Ziółkowska, Monika Różycka-Kosmalska, Tadeusz Pietras

**Affiliations:** 1Department of Clinical Pharmacology, Medical University of Lodz, 90-153 Lodz, Poland; 2Students’ Research Club, Department of Clinical Pharmacology, Medical University of Lodz, 90-153 Lodz, Poland; 3Department of Medical Biochemistry, Medical University of Lodz, 92-215 Lodz, Poland; 4Department of Clinical Electrocardiology, Medical University of Lodz, 92-213 Lodz, Poland

**Keywords:** non-alcoholic fatty liver disease, treatment, lifestyle intervention

## Abstract

Non-alcoholic fatty liver disease (NAFLD) is a serious health problem due to its high incidence and consequences. In view of the existing controversies, new therapeutic options for NAFLD are still being sought. Therefore, the aim of our review was to evaluate the recently published studies on the treatment of NAFLD patients. We searched for articles in the PubMed database using appropriate terms, including “non-alcoholic fatty liver disease”, “nonalcoholic fatty liver disease”, “NAFLD”, “diet”, “treatment”, “physical activity”, “supplementation”, “surgery”, “overture” and “guidelines”. One hundred forty-eight randomized clinical trials published from January 2020 to November 2022 were used for the final analysis. The results show significant benefits of NAFLD therapy associated with the use of not only the Mediterranean but also other types of diet (including low-calorie ketogenic, high-protein, anti-inflammatory and whole-grain diets), as well as enrichment with selected food products or supplements. Significant benefits in this group of patients are also associated with moderate aerobic physical training. The available therapeutic options indicate, above all, the usefulness of drugs related to weight reduction, as well as the reduction in insulin resistance or lipids level and drugs with anti-inflammatory or antioxidant properties. The usefulness of therapy with dulaglutide and the combination of tofogliflozin with pioglitazone should be emphasized. Based on the results of the latest research, the authors of this article suggest a revision of the therapeutic recommendations for NAFLD patients.

## 1. Introduction

Non-alcoholic fatty liver disease (NAFLD) is described as a fatty liver for which no secondary causes of the condition, such as alcohol use or hepatitis viral infection, can be found [1]. A diagnosis of this disease can be determined by identifying the connection between a lack of hepatocellular injury and the detection at least 5% fatty hepatocytes [2]. NAFLD, defined as a metabolic disease, is associated with a metabolic, biochemical and immune mechanism among diseases such as diabetes mellitus (DM), obesity, insulin resistance and dyslipidemia. This means that we can expect an increasing frequency of NAFLD cases, which is in line with the rising prevalence of obesity, diabetes and colonic diverticulosis [3,4]. According to this trend, there are indications regarding the role of genetics in the development of NAFLD [5]. It is accepted that NAFLD is a demonstration of a metabolic syndrome, the frequency of which is about 22% for women and 24% for men [6]. In reference to the latest reports, it was proposed that the term NAFLD should be changed to metabolic-associated fatty liver disease (MAFLD), which is broader and better corresponds with the metabolic links of this disease. IN regard to significant fibrosis of the liver, it was revealed that the MAFLD definition better correlates with this state than the NAFLD term does, and additionally, the MAFLD criteria are more useful for the identification of patients with liver fibrosis than the NAFLD criteria. Finally, it is important to note that the criteria for MAFLD diagnosis include the feature of liver steatosis and one of the subsequent following features: overweight or obesity, a DM2 diagnosis or evidence of metabolic dysregulation, in addition to a normal or low weight [7].

It is important to note that even if a patient is not overweight, they can develop NAFLD, which is proven by the fact that the prevalence of NAFLD in lean patients is 13.11% among the global population. However, metabolic dysfunction is associated with weight, and in lean patients, its prevalence is lower than it is in obese/overweight subjects. It was also revealed that metabolic syndrome, to a similar degree, correlates with liver fibrosis in both obese and non-obese patients, and when we took a closer look at MAFLD, we found that 20% of patients with this diagnosis are non-overweight [7,8].

NAFLD is the most common liver disease in the world [5]. The frequency of NAFLD in the general population is within the limits of 25 to 35%, while it was found to reach 70% within a group of patients with DM and obesity, and Moreover, we can observe the upward trend of NAFLD prevalence across the world [9,10,11].

NAFLD is known to be the main cause of the development of chronic liver disease. It leads to non-alcoholic steatohepatitis (NASH), cirrhosis and even hepatocellular carcinoma (HCC), and it is one of the main reasons for liver transplantation [9,10,12]. Progression to NASH occurs in 10 to 20% of patients with NAFLD [13]. In the case of HCC, less than 10% patients with NAFLD develop this complication, but studies have shown that NAFLD patients who received liver transplants were 10% more likely to be diagnosed with hepatocellular carcinoma than those without this diagnosis [14]. Non-obese people are less likely to develop NASH; the risk for this group is 3% [2]. Regarding cancers, there was a population-based cohort study which showed that patients with NAFLD have an increased risk of developing any cancer, such as colorectal, renal, bladder or uterine cancer; however, the main risk was HCC. It was proven in studies that the presence of NAFLD is connected with a higher frequency of cardiovascular events [15]. Regarding the cardiovascular diseases, we must note that it has been proven that NAFLD is connected with the development of cardiac arrhythmias and chronic kidney diseases (CKD) [16].

The treatment of NAFLD is problematic due to the lack of drugs accepted by the FDA in Europe. Although the guidelines recommend a narrow range of measures including lifestyle and diet changes, the use of hepatoprotective therapy is limited in NASH. The problem with this is that patients react differently to this treatment [9,17,18].

Although the European Association for the Study of the Liver point to the Mediterranean diet (MED), the most widely studied diet, as beneficial for patients with NAFLD, the most important recommendation is permanent weight loss, regardless of the type of diet [19]. Positive diet-related impacts on the patient condition include weight loss, an improved insulin resistance and a reduction in the amount of liver fat, which are known as factors which play roles in the NAFLD pathogenesis [9,12,14].

Despite the problem of NAFLD, it is still a challenge for modern medicine to develop pharmacotherapy for this disease.

## 2. Materials and Methods

The primary aim of this review is to summarize the results of different studies and compare them, allowing readers to determine the best treatment for NAFLD.

Following the Preferred Reporting Items for Systematic Reviews and Meta-Analyses (PRISMA) guidelines [20] (Appendix A), the PubMed Database was searched with a year-of-publication limit from January 2020 to November 2022, and for the limits regarding the type of study, we only referred to clinical trials and randomized controlled trials. The final search was performed on 10 December 2022.

A combination of Medical Subject Headings (MeSH) terms and specific search terms were used. We combined variations of following terms: “non-alcoholic fatty liver disease”, “nonalcoholic fatty liver disease”, “NAFLD”, “diet”, “treatment”, “physical activity”, “supplementation”, “surgery”, “overture” and “guidelines”.

The identified studies were downloaded and screened for inclusion based on prospectively defined inclusion criteria: English language studies referring to NAFLD treatment strategies, including pharmacotherapy, diet, supplementation, physical activity, surgeries and overtures. In the first screening, titles and abstracts or the full texts, if necessary, were reviewed. In the next step, the full text copies were reviewed. Information from the articles was extracted using a prospectively developed and tested template. The extracted information included the effects of the used treatment options, study populations and general information about the disease and general therapeutic options.

A total of nine hundred sixteen papers were identified with no duplicates as a first step. However, we excluded papers related to nonalcoholic steatohepatitis. According to the eligibility criteria, i.e., clinical trials and randomized controlled trials within the correct time of publication, one hundred forty-eight papers from the PubMed database were included in this study (Appendix A).

The results are reported in different paragraphs containing a short specification of each intervention and its results. All analyses were performed using Microsoft Office 365, CiteSpace (v.6.1. R2), and VOSviewer (v.1.6.18) to conduct bibliometric and visual analysis of all data.

## 3. Results

### 3.1. Diet and Supplementation

Body mass directly influences the majority of NAFLD cases. Due to this fact, patients are persuaded to change their diets and daily lifestyles. Often, a limitation of fats and carbohydrates needs to be accompanied by psychical activity. The recommended dietary intervention for NAFLD patients is plant-based diets and MED, whose efficiency is highly supported among researchers [19]. Dietary interventions include a fair share of NAFLD treatment due to weight reduction. In light of the fact that obesity is a known risk factor for hepatic steatosis, it has been observed that a 7–10% weight reduction decreases the level of steatosis in 88–100% patients [21,22]. Other authors suggest that even a 3% weight reduction has a beneficial impact on the liver fat amount, and in cases of liver inflammation and fibrosis, it has been proved that 5% weight loss results in the improvement of these parameters.

A reduction in body mass can be achieved with even a short follow up time for very-low-calorie diets. In addition, this kind of diet results in improvements of anthropometric measures, liver enzymes, blood pressure and lipid and glucose management. A positive impact on the state of patients with NAFLD was also observed for hypocaloric ketogenic diets (KD), despite the fact that they are supposed to contain a large amount of saturated fats, which can be explained by their poor carbohydrate content. The reasons for such an action can be observed in the caloric deficit. Moreover, it was revealed that lowering calories is a crucial dietary mechanism compared to the content of macronutrients, which leads to the improvement of hepatic steatosis [23].

As lifestyle changes can be difficult for patients, mobile apps that make it easier to maintain a diet and physical activity (PA) are emerging, and the effectiveness of such an app compared to standard medical controls was demonstrated in a recent study. Lim et al. revealed greater weight loss in patients using the nBuddy mobile app for 6 months in comparison to the standard controlled group [24].

The most popular type of lifestyle intervention, which is also the most effective for patients with NAFLD, is MED [9,21,25,26]. The positive impact of MED can be obtained even without calorie intake or body mass reductions [27,28]. The MED has been proven to reduce saturated fatty acid levels, which cause injury to the liver [21]. Furthermore, lifestyle interventions including the MED diet have an impact on DNA methylation, which is connected with reductions in HFC [25]. On the other hand, some authors could not prove the MED diet’s efficiency in reducing the liver lipid levels or demonstrate such effects using the homeostatic model assessment of insulin resistance (HOMA-IR), but visceral adipose tissue (VAT) reduction was observed in a 12-week follow up to MED [29]. In addition, a low-calorie diet can be used to handle NAFLD treatment due to its cardiometabolic change and weight reduction effects, even on the level of 5–7%.

MED modifications are also the subject of research, e.g., green MED, which is characterized by an increased intake of polyphenols and lower intake of red and processed meat. This subtype of MED, in combination with Mankai, green tea and walnuts, reduces the frequency of NAFLD by half. It was observed that the green MED diet has an advantage over the MED diet in terms of HFC reduction, but there was no difference in weight loss [30]. The other type of modification of the MED diet is the low-glycemic-index Mediterranean diet (LGIMD), in which less than 10% of the calorie intake originates from saturated fats, and the low glycemic index, connected with PA, resulted in a higher reduction in NAFLD than the control diet used in the cited study. The only diet advantage over PA is fact that the diet improves the state of hepatic damage markers [31].

The protein content in a diet, in addition to the hypocaloric diet and its impact on liver fat reduction, was the subject of research conducted by Xu et al. This study revealed that a diet with a high protein intake, described as more than 30% proteins, had a greater impact in decreasing the intrahepatic liver levels than diets with a low protein intake (10% protein) in a group of obese patients. Generally, both lifestyle interventions have good metabolic results, but the advantage of a higher-protein-intake diet is that it results in improvements in oxidative stress. In conclusion, in regard to protein intake, the results are clear: increasing the protein content in the diet with caloric restriction improves the decrease in the liver levels [32]. These data are in line with another study on hypocaloric and high-protein diets, which showed improvements through supplementation with β-cryptoxanthin for patients with NAFLD. At the endpoint, reductions in oxidative stress and inflammation were observed in the follow up to this intervention at 12 weeks [33].

It also was proven that a whole-grain diet has beneficial effects, including hepatic enzyme decrease and the improvement of liver steatosis. These effects were observed even after a 12-week follow up for this pattern of nutrition. In addition, this type of diet resulted in improvements of the intestinal microbiota state, level of glucose in serum fluctuations and blood pressure. Decreases in low-density lipoprotein (LDL) and total cholesterol (TC) were also observed [34]. The improvement of a reduction in liver fat was also observed after increased insoluble cereal fiber in the diet. In addition, this diet’s profitable effects on glucose maintenance, insulin resistance and the amount of uric acid were observed [35].

Regarding the dietary treatment approach of NAFLD, we can also specify the American Heart Association (AHA) and Fatty Liver in Obesity (FliO) hypocaloric diets. After 2 years of follow up, both resulted in weight loss and improvements in the body composition, fatty liver index and glycemic economy [36].

In response to the complications posed by NAFLD, researchers are searching for alternative solutions to address this problem. One of the new challenges is changes in the gut microbiota, observed in a group of patients with NAFLD. It was postulated that the pathological process of NAFLD is influenced by the intestinal microbiota, which creates the gut–liver axis. It was considered that it in this group of patients, the natural intestinal microbiota had a greater chance of reaching the small intestine, which leads to the impairment of the intestinal barrier and an increase in the concentration of endotoxins in the blood which, in turn, leads to hepatic inflammation and fibrosis. This research places the microbiota in view of a potential target for the treatment of NAFLD, and this may include the inclusion of MED or a low-carbohydrate diet, which has positive impacts on the gut microbiota and leads to an improvement in the fatty liver parameters [9,18,37].

It has been proven that omega-3 supplementation has beneficial effects on patients with NAFLD. Sangouni et al. proved that 12 weeks of supplementation with omega-3 at a dose of 2 g per day resulted in visceral fat reduction and improvements in the fatty liver index and lipid accumulation products among patients with DM [38]. Supplementation with omega-3 acids also reduced de novo lipogenesis (DNL) and improves fatty lipid oxidation. The amelioration of abovementioned processes could have a beneficial impact on NAFLD conditions [39]. It was also revealed that more than 3 months of supplementation with fish oils containing omega-3 PUFA resulted in alkaline phosphatase (ALP), liver fibrosis, alanine aminotransferase (ALT), triglyceride (TG) and fibroblast growth factor 21 (FGF21) reductions and adiponectin increase [40]. It was postulated that the FGF21–adiponectin axis could be a target through which fish oils could have an impact on NAFLD conditions [41]. The reduction in hepatic fibrosis may be connected with reductions in transforming growth factor beta (TGF-β) and tumor necrosis factor-alpha (TNF-α) levels, especially the former, which plays an important role in the progression from NAFLD to non-alcoholic steatohepatitis. There is a thesis stating that supplementation with omega-3 fatty acids and phytosterol ester, together, is more efficient in improving the NAFLD condition than supplementation with either of them alone [42].

It is known that for patients with NAFLD, the combination of lifestyle changes and probiotic or prebiotic supplementation is more beneficial in improving the glycemic economy and leptin levels than lifestyle interventions alone [43]. Accordingly, the influence of probiotic supplementation has become the basis of many studies.

Supplementation with multi-strain probiotics (MCP^®^ BCMC^®^ strains), including Lactobacillus and Bifidobacterium species, in NAFLD patients limited the permeability of the intestinal membrane, which means that probiotics may play an auxiliary role in the treatment of NAFLD [37]. The effectiveness of this symbiosis has been proven only in terms of changes in the intestinal microflora [44]. A study of supplementation with VSL#3^®^ revealed that among patients with NAFLD, this type of probiotic supplementation does not decrease the cardiovascular risk, insulin resistance or probability of liver injury. It is important to note the fact that the small sample size of this study was a limitation. Regarding intestinal bacteria, some researchers went a step further and explored the influence of fecal microbiota transplantation on patients with NAFLD. Patients were instructed to follow a healthy diet and perform over 40 min of physical activity. The results showed that this intervention ameliorated gut dysbiosis, which resulted in the improvement of the liver fat state. It is interesting to note that a better improvement was observed in patients with a BMI lower than 25 compared to patients with a BMI of at least 25 or more [45].

It is known that development of NAFLD is connected with low vitamin D levels, which means that supplementation with this constituent could be a target for the treatment of the aforementioned disease [46]. Due to the fact that insulin resistance plays a role in the pathogenesis of NAFLD, attempt are being made to identify a target for the treatment of this disease. Recent studies showed a positive effect in the reduction in insulin resistance and improvement of insulin plasma levels through supplementation with calcitriol together with lifestyle modifications. Regarding vitamin D’s effect in improving the liver state, the research data are contradictory [46,47]. Vitamin D supplementation is efficient in reducing controlled attenuation parameters (CAP) and liver stiffness measurements (LSM) which, respectively, reflect liver steatosis in fibrosis. Reductions in the above parameters were achieved due through 1 year of vitamin D supplementation at a dose of 1000 IU per day in patients with an NAFLD diagnosis [48]. The efficiency of vitamin D supplementation was also proven in a study which assessed the action of 6 weeks of daily supplementation with 2000 IU vitamin D per day with a comparison between the beginning and the end of the study, together with exhaustive eccentric exercise (EEE). At the endpoint, improvements in the anthropometrics measures and lipid profile and a decrease in the level of liver enzymes were observed. Vitamin D could improve alterations which may be a result of EEE, such as a decrease in HDL or increase in the liver enzymes [49].

There are many studies confirming the therapeutical effectiveness of plant and animal derivatives in NAFLD patients. The majority of the results show that plant derivatives (and propolis, that is, animal derivatives) improve anthropological parameters, the lipid profile, hepatic steatosis and anti-inflammatory and/or antioxidant factors. The studies of natural compounds that have been conducted on NAFLD patients are presented in Table 1.

According to studies conducted on natural extracts influencing and improving the NAFLD state, one must mention Livogen Plus^®^, which is a nutraceutic composed, among other ingredients, of bergamot polyphenolic fraction, curcumin complex, omega-3 PUFAs, artichoke leaf extract, black seed oil from Nigella sativa, pricoliv, which is a substance sourced from *Picrorhiza kurroa*, glutathione, S-adenosyl-l-methionine and other natural ingredients. The abovementioned substances have been proven to have some beneficial effects on different pathways in one or more of the following: metabolism, inflammation, glycemic and lipid maintenance and the hepatic state. A 12-week follow up to supplementation with Livogen Plus^®^ at a dose of six capsules per day resulted in reductions in liver steatosis in 62% of patients. Moreover, improvements in insulin resistance and decreases in diastolic blood pressure and oxidative stress, demonstrated by an increase in biological antioxidant potential, were observed [76]. Efficiency in hepatic fat reduction was also proven for another nutraceutical administered through 12-week supplementation, containing extracts from *Bergamot* polyphenolic fraction and *Cynara Cardunculus*, at a dose of 300 mg per day [77].

According to the roles of oxidative stress and insulin resistance in NAFLD development, researchers explored substances which alleviate both of the aforementioned factors in depth. A period of 12 weeks on a hypocaloric diet with 30 mg per day of elemental zinc supplementation, at the endpoint, resulted in a decrease in insulin resistance and improvement of the insulin serum superoxide dismutase 1 and malondialdehyde levels in overweight NAFLD-diagnosed patients. This study proved the antioxidative and insulin-resistance-enhancing efficiency of zinc, which shows that its supplementation could be a supportive therapy for NAFLD [78]. Another study determined the efficiency of 12-week supplementation with zinc gluconate at a dose of 220 mg per day, which is equal to 30 mg of elemental zinc, in connection with a calorie-restricted diet. At the endpoint, reductions in ALT, GGT and WC were observed. [79]. Another substance which has beneficial effects on oxidative stress and inflammatory processes is saffron. Its supplementation was examined over a 12-week follow up at a dose of 100 mg per day [80].

It is proven that chromium has beneficial effects on liver injury protection, insulin resistance and lipid management. This observation became the starting point for a study which proved that 12-week supplementation with chromium picolinate at a dose of 400 μg per day resulted in decreases in the atherogenic plasma index, TG levels, insulin resistance, IL-6, TNF-α, C-reactive protein (CRP) and fetuin-A in patients with NAFLD. Fetuin-A is a glycoprotein synthesized in the liver which is known to be a marker of NAFLD. On the other hand, an improvement in the liver fat amount or in lipids other than TG has not been observed [81]. The studies of dietary interventions that have been conducted on NAFLD patients are presented in Table 2.

### 3.2. Physical Activity

As stated above, most studies connected the impact of diet on NAFLD with physical activity. The necessity of introducing physical activity and its effectiveness in the prevention of the development of liver steatosis in the treatment of patients with metabolic disorders was confirmed by a study conducted on overweight children followed up to 22 weeks so as to assess their cardiorespiratory fitness. As a result of this intervention, decreases in the liver fat amount and GGT and the improvement of the AST/ALT and TG/HDL ratios were observed at the endpoint of the study. Moreover, insulin resistance improvement was demonstrated, but it was not statistically significant. Based on the fact that cardiorespiratory fitness can reduce the cardiometabolic risk, it is a good measure to introduce this intervention into the treatment of patients with NAFLD [83]. Although PA offers the best results in reducing the amount of fat in the liver while reducing the body weight, even without it, the efficiency of this measure was proven. Additionally, regular PA improves muscle strength and general physical fitness, which have positive impacts on the patient’s state [84]. Moreover, regular PA is recognized as a prevention method for metabolic factors, among which NAFLD is included, and the development of cardiovascular complications. It is known that physical activity is part of lifestyle changes, which represent the primary treatment for NAFLD. It was revealed that aerobic exercises with LGIMD resulted in the greatest improvement of the state of a group of patients with NAFLD, but combined resistance exercise programs also show great results in improving the patient’s state. It is known that resistance activities such as push-ups or squats have effects on metabolic syndrome and lead to insulin sensitivity improvement and ALT reduction, and decreases in the cytokeratin 18 (CK18) and FGF21 levels were observed after applying this type of PA. Based on the fact that CK18 is a marker of hepatocyte apoptosis and that FGF21 could be used as a marker of liver steatosis, this type of physical activity could act as method of prevention of NAFLD progression [85].

Aerobic exercise leads to reductions in BMI and insulin resistance, which shows its advantage over resistance activity, which only results in the improvement of insulin resistance. This positive aspect of PA was observed even without connection to a diet [31,86]. According to above mentioned data, comparing between aerobic and resistance activity, some researchers stated that both provide similar results in reducing liver and visceral fat. It is proven that PA for at least 3 days per week results in an improvement of the NAFLD state among patients, and the greater the frequency of exercise is, the better the outcomes will be. Another study proved that 8 weeks of aerobic exercises for 165 min/week was efficient in reducing liver steatosis and fibrosis, effects which could be enhanced by whole-body vibration training and improvement of the transaminase levels, but in this aspect, a better effect was shown in groups followed up with aerobic exercises only. On the other hand, the enhancement of insulin resistance was shown to be significant only in the group followed up with aerobic and whole-body vibration training. In addition, it was proved that physical activity has an impact in enhancing the quality of life [87]. A study was conducted wherein older women with NAFLD were followed up to 12 weeks, performing non-linear resistance training with or without curcumin supplementation. At the endpoint, in both cases, reductions in ALT and AST were observed, which confirm this measure’s beneficial effect on the state of patients with NAFLD and shows that this type of intervention could be helpful in managing NAFLD [64].

Moderate-intensity exercise is the most widely promoted form due to the ability of patients to perform it, despite the fact that it is less effective than heavy training [88]. Beneficial reductions in ALT, AST, CRP and TG were reached by aerobic interval training 3 days a week for 6 weeks [89]. Insulin resistance improvement was also proven for a hybrid training system defined as physical activity combined with neuromuscular electrical stimulation. A regime of 30 min of treadmill exercise per day performed 3 times a week with the aforementioned stimulation resulted in insulin resistance enhancement and decreased levels of IL-6 and SeP, a hepatokine whose level is elevated in subjects with NAFLD or diabetes. The higher decline of IL-6 level was observed in group followed up to electrostimulation in comparison to physical activity only [90].

It has been proved that the introduction of PA entails changes in DNA methylation [25]. The positive influence of physical exercise was also revealed to be a factor which has a positive impact on the microbiota, which could be a result of a decrease in secondary bile acids released by intestinal bacteria [9,18,85]. Better body mass and intrahepatic fat reductions were reached by combining PA and diet in a comparison of these individual interventions [31].

High-intensity interval training (HIIT) performed by patients with NAFLD without dieting resulted in reductions in the plasma glucose level and WC, even without a reduction in weight. Regarding insulin resistance after HIIT, researchers did not observe changes in comparison to the control group. A reduction in WC carries a decline in general mortality and the risk of developing DM2. It is accepted that this intervention results in hyperglycemia protection, a decrease in bile acids in the plasma and risk of hepatic disorders, such as fibrosis [86]. The above data are in line, concerning HIIT’s efficiency, with Abdelbasset et al., who carried out a comparison between HIIT and moderate-intensity continuous aerobic exercise, and no differences between these two groups were observed in the outcomes of the study. In addition, it was proven that both forms of exercise have beneficial effects on the hepatic and metabolic states [91]. These data are in line with another study, which proved that, over a 16-week follow up, moderate PA efficiency resulted in reductions in liver steatosis, weight, BMI, levels of fasting glucose and ALT and LDL-C in the plasma. The intensity of training in this research was set to 4 to 5 days per week for 20 min to one hour, increasing with the duration of the test [92].

### 3.3. Pharmacotherapy

The multifaceted nature of the causes of NAFLD development and the relationship between this disease and other disease entities, such as diabetes, metabolic syndrome, diverticulitis or chronic inflammation, are the basis of the need for a holistic approach to the patient’s condition. In addition, this provides opportunities for treatment options due to the fact that an improvement in one of the abovementioned conditions may result in an improvement in the others [93].

According to fact that systemic inflammation and oxidative stress worsen the prognosis of NAFLD, treating these conditions may be carried out in connection with NAFLD treatment, and due to this fact, hydroxytyrosol and vitamin E and their antioxidant activities have been tested in child patients. There are data showing that 400 IU of daily vitamin E treatment resulted in a decrease in the ALT, AST and GGT levels [94]. It was proven that a 4-month combined treatment of vitamin E and hydroxytyrosol resulted in improvements in insulin resistance and the hepatic fat amount due to the beneficial effect on the plasma levels of inflammatory factors [95,96]. It should be noted that vitamin E is not effective in all patients, but it can be helpful, and it is worth trying to include it in therapy due to the potential beneficial effects that can be achieved. According to these findings, more studies on vitamin E with different doses and patients groups are needed to prove its efficiency in managing the liver state [97,98]. The oxidation process of fat could be a focal point of NAFLD treatment, as was proven by Zeybel et al. They introduced a combined metabolic activator (CMA) consisting of L-carnitine tartrate, L-serine, N-acetyl-L-cysteine and nicotinamide riboside for 10 weeks to follow up patients with an NAFLD diagnosis. Upon comparison with the placebo group, they proved that the CMA treatment benefits included a reduction in oxidative stress and improvement of mitochondrial functions and liver fat oxidation, which were not connected with weight reduction. This resulted in benefits for the microbiomes of the intestine and oral cavity, as well as a reduction in the liver fat amount and liver enzymes in the serum. An improvement of the hepatic inflammatory process and reduction in the plasma levels of creatinine and uric acid were also observed. Uric acid is known to play a role in the development of liver steatosis; thus, its reduction was beneficial. It is important to note that CMA therapy is proven to be safe for patients, and its toleration by patients is good [99]. On the other hand, some researchers did not prove the action of L-carnitine in improving the state of patients with NAFLD. L-carnitine’s inefficiency was investigated in children, adolescents and patients with polycystic ovarian syndrome [100,101]. Due to these results, more research on L-carnitine and its connections with other substances are needed to prove its efficiency. Recently, researchers explored endogenous metabolic modulators (EMM) as potential drugs for NAFLD treatment. The results of the 16-week intervention with AXA1125 or AXA1957, each of which belongs to the EMM group, were satisfactory. Both of the above were safe and well tolerated by patients. At the endpoint, reductions in ALT, liver steatosis, serum markers of hepatic fibrosis, insulin resistance and radiological markers of fibrosis and inflammatory process were observed. In each of the above aspects, AXA1125 was more efficient in comparison to AXA1957. Moreover, the first reduced the level of the apoptosis marker K-18 M65, while the second caused this marker to increase. This study sheds light on new treatments for NAFLD using EMM [102].

As is well known, the connection between glucose management and the development of NAFLD has been proven. Due to this fact, the potential solution for NAFLD treatment is an improvement in this matter [103]. Hyperglycemia treatment includes many classes of drugs, one of which is sodium-glucose cotransporter 2 (SGLT2) inhibitors, which were proven to have pleiotropic effects expressed as a reduction in cardiovascular risk [104]. Moreover, regarding SGLT2 inhibitors, their benefits in NAFLD treatment for patients with DM2 have been proven by numerous studies, which have shown that SGLT2 inhibitors enhance glycemic control, with reductions in the visceral fat amount and weight, which can improve liver steatosis [104,105,106,107,108,109]. Other authors proved a reduction in the liver enzyme levels in the plasma with dapagliflozin and ipragliflozin [103,105]. Moreover, ipragliflozin, tofogliflozin and empagliflozin treatments have been proven to reduce hepatic fibrosis. On the other hand, for ipragliflozin treatment, a reduction in the liver fat amount was not observed, but this action was shown to result from empagliflozin, tofogliflozin and dapagliflozin treatment [104,105,107,108]. Most researchers verified the efficiency of SGLT2 inhibitors in patients with comorbidities of DM2 and NAFLD, but for empagliflozin, the ability to improve the hepatic state was proven in patients with NAFLD but without diabetes [107]. Studies have also demonstrated the improvement of NAFLD due to pioglitazone treatment, and it is known that this medication improves insulin resistance and lowers the blood glucose level. After pioglitazone treatment, an indirect improvement in liver steatosis and inflammation was observed [110]. Recently, it was discovered that women respond better to pioglitazone treatment than men [111]. Regarding the beneficial effect of a reduction in liver fat, it is understood that this is caused by an increase in the adiponectin levels due to pioglitazone [103]. It has also been shown that pioglitazone therapy is associated with a better effect in reducing the hepatic fat and improving fat metabolism compared to tofogliflozin. However, it should be emphasized that, unlike flozin, the use of pioglitazone may lead to weight gain [104]. On the other hand, some researchers supported the notion that dapagliflozin is more effective than pioglitazone in NAFLD treatment [109]. This was the starting point of a study which tested the efficiency of a combination of pioglitazone and tofogliflozin in NAFLD treatment. This study revealed that the combination of the above results was more effective in improving the patient’s state, with greater safety of use than each of them independently [112].

The other group of antidiabetic drugs is dipeptidyl peptidase-4 inhibitors (DPP-4I), whose mechanism of action is connected with the maintenance of incretin hormones. It is known that DPP-4I results in an increase in the level of exogenous glucagon-like peptide-1 in the plasma, which entails an increase in glucose levels in the plasma. One of medications which belongs to this group is sitagliptin. It is known that it has a beneficial effect on diabetes control, which was proven by Wang et al. through reductions in HbA1c and FPG in a 24-week follow up to sitagliptin treatment at a dose of 100 mg per da. At the endpoint, a reduction in liver fat was observed but was not statistically significant, probably due to the short time of the treatment and small number of participants in this study. This potential beneficial effect needs to be verified in future research based on more respondents [113]. To support the need for research on the effects of DPP-4I on liver health, a study showed that evogliptin has beneficial effects in the improvement of inflammatory signaling pathways in isolated liver cells, connected with its modulation of autophagy. The above data show the potential efficiency of this group of medications in hepatic fibrosis and inflammation treatment [114].

Exenatide belongs to the glucagon-like peptide-1 receptor agonist (GLP-1RA) group and is used in diabetes mellitus type 2 treatment. Researchers examined exenatide’s effects on liver health. A 24-week treatment with exenatide was conducted as follows: 5 μg twice daily for 4 weeks and 10 μg twice daily for the remaining time, which was introduced into patients with newly diagnosed DM2. The treatment resulted in improvements in glycemic control, the lipid profile, anthropometric parameters and visceral and subcutaneous fat content, with reductions in GGT, AST and ALT, as well as hepatic steatosis and the fibrosis index [115]. According to the fact that another agent, GLP-1RA, impacts on NAFLD treatment, studies were conducted. Semaglutide, in a 72-week treatment at a dose of 0.4 g per day, resulted in weight reduction, improvements in glycemic maintenance and the plasma level of hepatic enzymes, and a reduction in liver fat content [116]. A 24-week treatment with another substance from this group, dulaglutide, resulted in the improvement of GGTP and a reduction in the liver fat amount in a group of patients with DM2 and NAFLD [116]. It is known that liraglutide, a GLP-1RA treatment, leads to a reduction in hepatic fat and the improvement of the lipid profile and liver function parameters [117,118]. Another study proved the efficiency of liraglutide in reducing liver steatosis and the fetuin-A level, which is linked to the hepatic fat content and body weight, after 24 weeks of treatment [119]. The positive effect of liraglutide co-treatment with metformin in obese diabetic patients with NAFLD was also proven by Guo et al. This research proved the efficiency of this composition in reducing liver steatosis, ALT, AST, insulin resistance factor and visceral and abdominal fat [120].

A combination of dapagliflozin, an SGLT2 inhibitor, and exenatide, a GLP-1RA inhibitor, was tested by Harreiter et al. A 24-week follow up of this intervention did not prove the beneficial effects of this composition in reducing the hepatic fat amount or liver enzymes in comparison to dapagliflozin treatment alone. It only showed that with this composition, better glycaemic control, measured by the lowering of HBA1C and fasting glucose level, was reached in comparison to the dapagliflozin alone group. Regarding weight reduction and the hepatic state, the outcome was similar between the groups who received the composition and SGLT2 inhibitor alone. This study also proved that hepatocellular lipid reduction is connected with weight loss, the improvement of anthropometric aspects and abdominal and visceral fat changes. A correlation between better glycemic control and a reduction in hepatocellular lipids was not observed, but the remission of DM2 may be connected with a reduction in the abovementioned factor, as other studies have proven [121]. On the other hand, a post hoc analysis in a multicenter, double-blind, randomized, active-controlled, phase 3 study named DURATION-8 proved the efficiency of this co-treatment in reducing liver steatosis and fibrosis biomarkers in 695 patients. The outcomes were measured after a 52-week follow up. It was also proven that co-treatment with abovementioned medications is more beneficial than treatment with either of them alone [122].

Insulin continues to be an important part of diabetes management. Based on the high co-morbidity of diabetes and NAFLD and the pathways which connect these diseases, the impact of insulin treatment is also under investigation as potentially yielding NAFLD treatment benefits. A 24-week glargine treatment in newly diagnosed diabetes mellitus type 2 patients with NAFLD diagnoses resulted in reductions in liver steatosis, TG, free fatty acid (FFA), WC, ALT and GGT and improved glucose management control, measured by the improvement of the HbA1c level. In this study, a comparison between exenatide and glargine in NAFLD treatment was performed, and the outcomes showed that exenatide is more effective in improving anthropometric parameters and reducing LDL-c and fibrosis of the liver [115]. Guo et al. also verified the efficiency of insulin glargine combined with metformin treatment and showed improvements in the liver, abdominal and visceral fat resulting from this treatment. It is important to note that the co-treatment composed of liraglutide and metformin was more efficient in producing these changes than metformin and glargine during follow up [120].

The combination of ezetimibe and rosuvastatin was postulated as a treatment that could lead to hepatic steatosis reduction, which was proved by Cho et al. in a study which introduced this kind of treatment into patients for 24 weeks. Moreover, it was proven that the combination of these two medications is safe for patients and more effective than rosuvastatin therapy alone [123]. As we are concerned with the treatment of lipid disorders, it is worth mentioning pemafibrate’s efficiency in reducing liver stiffness. This substance belongs to a selective peroxisome-proliferator-activated receptor (PPAR) α modulator group, which is used in hypertriglyceridemia treatment. In addition, an improvement of hepatic serum enzymes and reductions in LDL-C and TG after using this treatment for 72 weeks were observed [124]. There is also saroglitazar, which belongs to the PPAR-α/γ agonists group, whose efficiency in the treatment of NAFLD was proven. A 16-week follow up to saroglitazar treatment at a dose 4 mg per day revealed its beneficial effects in reducing ALT, insulin resistance, atherogenic dyslipidemia and the hepatic fat amount. It is understood that due to lipoprotein maintenance enhancement, this treatment decreases the cardiovascular risk. In addition, saroglitazar improves bile acid metabolism, the dysregulation of which is known to be a factor in NAFLD development [125].

Ursodeoxycholic acid (UDCA) is postulated to have hepatoprotective and antioxidant impacts. It was revealed that UDCA treatment resulted in a decrease in AST, ALT and GGTP. Moreover, a reduction in the miR-122 level was observed. MiR-122 is known to be highly specific to the liver and can be used as a factor related to drug-induced liver damage; hence, a decrease in its level could be a sign of the hepatoprotective efficiency of UDCA [94]. Another study proved that 6-month treatment with 15 mg/kg UDCA per day shows efficiency in the enhancement of ALT, AST, GGT and the lipid profile and a reduction in liver fat amount. The results achieved in this study were independent of body mass reduction. Moreover, a reduction in the 10-year atherosclerotic cardiovascular disease risk was observed for the women in the study group, and for both men and women, an improvement in carotid intima-media thickness was observed [126].

According to fact that low levels of growth hormone in the serum may play a role in NAFLD development, patient performance in this aspect was tested [127]. Pan et al. revealed the efficiency of the recombinant human growth hormone (rhGH) somatropin in reducing the liver fat amount. In this study, 24 weeks of rhGH therapy in group of young obese people with NAFLD diagnoses resulted in a 36% relative reduction in liver steatosis, without glycaemia increase or other significant side effects. The adverse effect of a reduction in weight in this study may not be connected with rhGH treatment but may be a result of lifestyle changes; thus, a study on a larger group of patients is needed [128]. These data are partially in line with those of another study on rhGH’s impact on obese boys aged 8–16 with NAFLD diagnoses but without diagnoses of organic diseases of the hypothalamus or hypophysis. After a 6-month follow up to rhGH administration, an increase in HDL-C and reductions in the LDL-C, GGT, AST, ALT and CRP levels were observed. On the other hand, a statistically significant improvement of liver health was not observed by ultrasonography. In addition, rhGH was viewed as a potential agent for improving cardiovascular and metabolic complications, which are connected to obesity in patients with NAFLD [127].

Improvements in the anthropometric parameters of patients with an NAFLD diagnosis were observed following melatonin treatment at a dose of 6 mg per day for 12 weeks. Melatonin’s efficiency was also proven through a decrease in the grade of fatty liver, ALT, AST and hs-CRP. Moreover, reductions in SBP and diastolic blood pressure (DBP) and an increase in the leptin serum level were proven to result from this treatment. According to the abovementioned parameters’ improvement, melatonin could be a good management option for NAFLD treatment [129].

A beneficial effect on the state of NAFLD-diagnosed patients is known to be characteristic of fatty acid synthase (FASN) inhibitors. One of these medications is a β-ketoacyl reductase, TVB-2640, which was proven to improve inflammatory, biochemical, metabolic and fibrotic biomarkers and reduce hepatic steatosis. These effects were results of a 6-week treatment with 300 mg of this substance [130]. It is also known that the main factor in NAFLD development in subjects with insulin resistance is de novo lipogenesis, which is the focal point of the action of this drug [131]. A study was conducted which, at the endpoint, showed reductions in diastolic blood pressure, liver steatosis, total cholesterol, LDL-c and ALT following 10 days of treatment with TVB-2640, even at a dose of 100 mg per day, in patients with metabolic syndrome. For the other tested doses (50 mg and 150 mg), significant changes in total cholesterol and LDL-c were not observed. HDL-c level decrease in the serum was observed in patients who were followed up after treatment with 50 and 100 mg of this substance. In this study, de novo lipogenesis was reduced in subjects treated with 100 and 150 mg of TVB-2640, while 50 mg did not yield this result. The best outcomes, a reduction in intrahepatic TG and other mentioned benefits, were observed in the group treated with 100 mg of the medication [131]. Effectiveness in reducing the liver fat amount and hepatic de novo lipogenesis was also proven for another FASN inhibitor, FT-4101, administered at a dose of 3 mg per day for 12 weeks of treatment [132]. Both of the above were proven to be safe for use [130,132].

It is known that a decrease in adenosine-monophosphate-activated protein kinase (AMPK) activity is the result of impaired regulation of metabolism. AMPK is responsible for modulating metabolic and inflammatory processes such as the stimulation of glucose uptake by cells and fatty acid oxidation and plays a major role in de novo lipogenesis inhibition. These properties became the starting point of a study on a drug that activates AMPK, PXL770, administered at a dose of 500 mg per day over 4 weeks of follow up. At the endpoint, the safety of PXL770 for use was proven, and the authors also observed improvements in insulin sensitivity and glycemic parameters, the inhibition of fructose-stimulated de novo lipogenesis and reductions in GGT and some of the diacylglycerol and triacylglycerol subdivisions. The above data indicate the positive metabolic effect of PXL770 treatment and its potential as a therapeutic strategy for NAFLD. However, more research is required to prove its efficiency in a larger group of participants [133].

According to enzyme reticence, the inhibition of ketohexokinase (KHK) has become a focal point for NAFLD treatment. This enzyme plays an important role in fructose metabolism, and its level is elevated in obese patients with NAFLD. Additionally, it is known that high fructose consumption leads to metabolic dysregulation, which may cause the development of NAFLD [134]. Research was conducted in which patients were followed up over 6 weeks of daily treatment with 300 mg of PF-06835919, which is known as a KHK inhibitor. At the endpoint, the safety of this treatment, with reductions in hepatic steatosis and the liver enzymes, were observed. Moreover, its beneficial effects on cardiometabolic factors were proven in other studies [135].

The action of low-molecular fucoidan (LMF) in connection with high-stability fucoxanthin (HSFx) on the NAFLD state was verified by Shih et al. The patients underwent 24-week treatment with 825 mg of LMF and 825 mg of HSFx tablets p.o. twice a day. At the endpoint, reductions in inflammation, lipotoxicity, the liver fat amount and insulin resistance were observed. The authors believed that for patients with NAFLD, this kind of treatment could be useful for alleviating hepatitis and liver fibrosis [136].

There are reports of lubiprostone’s beneficial effects in NAFLD treatment. Lubiprostone is a laxative medication which is known to enhance gut permeability and is used in constipation treatment. On this basis, a study was conducted on patients with constipation and NAFLD diagnoses. Based on the results, it was proven that lubiprostone treatment for 12 weeks reduced the levels of liver enzymes and hepatic fat content. Endotoxemia improvement was also observed. A better ALT reduction was observed in the groups who received a higher dose lubiprostone (24 μg vs. 12 μg). On the other hand, patients who were followed up after a greater dosage often reported adverse effects such as diarrhea, vomiting and nausea. More research is needed on the effects of lubiprostone in groups of patients without constipation [137].

According to fact that NAFLD development is identified as a side effect of pancreatoduodenectomy, methods of prevention are wanted. One of these methods could be the high-dose administration of digestive enzymes, which was proven to reduce the initial state of NAFLD when compared to normal doses of the enzyme. The preventive effect of this supplementation could be exploited as a protective factor for patients with exocrine pancreatic disability and undergoing pancreatoduodenectomy [138].

The rebate of AST, ALT and liver fatness was observed following an A3 adenosine receptor (A3AR) agonist treatment named namodenoson. Moreover, the anti-inflammatory efficiency of this treatment was proven by its effect in increasing the level of adiponectin. In addition, preclinical studies have proven the cardioprotective and neuroprotective actions of this substance. Namodenoson is known to be safe and could be a new kind of treatment for patients with NAFLD [139].

Hydrogen is known to have an anti-inflammatory action and plays a role in the regulation of the autophagy process. Based on the fact that the inflammatory process plays a role in NAFLD development, researchers decided to verify the effectiveness of inhalations with a mixture composed of hydrogen and oxygen in the treatment of this disease. A 13-week follow up to one-hour inhalation of a mixture containing 66% hydrogen and 33% oxygen at a flow rate of 3 L per minute per day resulted in improved liver steatosis and the promotion of the liver autophagy process, which can be described as a protective factor [17].

Beneficial effects in lowering the serum levels of ALT and GGT and improving insulin sensitivity were proven for 4-week treatment with PX-104 at a dose of 5 mg daily in patients with NAFLD diagnoses without diabetes mellitus. PX-104 is a non-steroidal farnesoid X receptor agonist which plays a role in the regulation of glucose, lipid and bile acid maintenance. Moreover, its potential enhancement of the gut microbiota and reduction in the bile acid level in stool were observed. According to the fact that lipid profile, BMI and liver fat amount improvements were not observed, further studies on this medication are needed to confirm its aforementioned beneficial effects [140].

The efficiency of n-3/omega-3 fatty acids in influencing the NAFLD condition was mentioned in the Diet and Supplementation subsection. For this reason, researchers are working on these kinds of supplements so as to introduce them into treatment. One of these treatment options is epeleuton, which is a novel synthetic second-generation n-3 fatty acid. In NAFLD patients, a 16-week treatment with 1 or 2 g of epeleuton daily resulted in dose-dependent reductions in the TG and total cholesterol levels. Moreover, improvements in insulin resistance, glucose, HbA1c and markers of inflammation and the endothelial dysfunction levels were observed, representing a potential reduction in cardiovascular risk. Reductions in liver steatosis and the ALT levels, without statistical significance, were observed following this treatment [141].

Lastly, a new NAFLD treatment opportunity has emerged in the form of IONIS-DGAT2Rx, which is an antisense oligonucleotide inhibitor of diacylglycerol-o-acyltransferase 2 (DGAT2). This enzyme is responsible for catalyzing the final step of TG synthesis. A 13-week follow up to this treatment s.c. at a dose of 250 mg once per week resulted in a reduction in hepatic fat. Significantly, 48% of patients reached at least 30% fat reduction. This substance could be an excellent treatment option for patients with NAFLD diagnoses [142].

Novelties in the pathogeneses of many diseases, including NAFLD, can be seen in epigenetic alterations in microRNAs (mi-R). It is known that some types of microRNAs may play a role in the liver condition; thus, they are known to be regulators of oxidative stress, the inflammatory process and lipid metabolism. As examples of microRNAs connected with NAFLD, we can mention, for example, miR-16, miR-2, miR-155 and miR-223. MiRNAs exhibit multiple functions depending on their type, and it was discovered that miR-223 could reduce liver fibrosis due to its antifibrotic effect. Future research on the impacts of miRNAs is needed in order to better understand their functions and reveal the potential beneficial effects of their actions. MicroRNAs are a promising target for the development of therapies for many diseases, including NAFLD [67,143].

Today, the achievements of traditional medicine are often discussed, and this also applies to the treatment of NAFLD. Wang et al. proved the effectiveness of traditional Chinese medicine in the treatment of NAFLD. In this study, researchers closely examined Danshao Shugan granules (DSSG), which, in traditional Chinese medicine, are used for liver protection. The effect of using DSSG is beneficial in improving liver parameters and lipid profile. In addition, in this study, DSSG were proven to be more effective in NAFLD treatment than rosiglitazone, but the latter was more beneficial in reducing the glucose level in plasma. It is important to note that DSSG therapy has fewer side effects in comparison to the standard drugs used [144]. The studies on the efficiency of the chosen treatment options conducted on NAFLD patients are presented in Table 3.

### 3.4. Surgeries and Overtures

Recently, non-invasive focused ultrasound cavitation connected with beneficial results of aerobic activity was proven to result from NAFLD treatment. Researchers observed weight, BMI, WC and visceral and liver fat reductions following this intervention. This procedure appears to be safe for patients, and in the study, the patients did not complain of ailments due to non-invasive focused ultrasound cavitation [145].

Favorable outcomes were achieved following hydrothermal duodenal mucosal resurfacing (HDMR) in patients with DM2 diagnoses. This treatment is based on endoscopic duodenal circumferential mucosal lift and the ablation of the mucosa and leads to improved glycemic control, a result which is in line with the length of the fragment which undergoes the procedure. In REVITA-2 study outcomes for this procedure are satisfying. After this intervention, beneficial effects on glycemic control and a decrease in liver steatosis were observed. More benefits of HbA1c in this study were experienced by patients with higher fasting plasma glucose levels at the beginning of the study, measured as at least 10 mmol/L, in comparison to patients with lower levels of this parameter [146].

Bariatric surgeries are proposed to reduce weight in obese patients. Due to the connection between obesity and NAFLD development, researchers explored this kind of surgery in NAFLD treatment. It is known that the levels of hepatocyte-derived extracellular vesicles reflect the liver fibrosis and steatosis state. A decrease in this marker is observed after weight loss surgeries, which may reflect the improvement of the NAFLD state [147].

A comparison of two types of bariatric surgeries, sleeve gastrectomy (SG) and Roux-en-Y gastric bypass (RYGB), was conducted. At the endpoint, the advantage of SG in improving the liver fat amount and fibrosis was proved in comparison to the second surgery. However, on the other hand, some studies say that in a long time RYGB is more beneficial in remission of comorbidities than SG [148].

Lastly, the efficiency of electroacupuncture treatment was proven in NAFLD. Acupuncture is a kind of overture derived from traditional Chinese medicine. In Draz et al.’s study, the modification of this treatment was performed. In electroacupuncture, small needles with electrical potential are connected to a stimulator. In this study, the following acupuncture points were used: liver 3, gall bladder 34 and stomach 36. The follow up to this intervention, administered three times a week for six weeks, resulted in reductions in TG, CRP, AST and ALT. Moreover, even if aerobic interval training led to reductions in the same parameters as those observed in the comparison group in this study, the benefits of electroacupuncture were better [89]. The above data are in line with another study which verified electroacupuncture’s effectiveness in NAFLD. After a 12-week follow up to this intervention, administered 3 times a week in connection with lifestyle control, reductions in liver steatosis, AST, ALT and GGT were observed. Additionally, improvements in the anthropometric measurements, lipid profile and glucose economy, including the fasting plasma glucose level, and insulin resistance decrease were obtained [149].

## 4. Conclusions

NAFLD, despite its global prevalence, is still a therapeutic challenge for modern medicine. New therapeutic solutions are needed to improve the quality of life of patients with this diagnosis and prevent the development and complications of this disease. Lifestyle modifications are still the basis of treatment, and as we mentioned here, they are effective in managing metabolic syndrome, reducing the liver fat content and preventing the deterioration of the patient’s condition. However, they may be difficult to introduce and maintain; hence, other treatment options are needed. We can see that the interventions that are successful in NAFLD treatment range from traditional methods to the latest drugs that are still under study, ranging from dietary supplementation to surgeries. It is important to adapt the methods to each patient’s needs so as to offer him/her the greatest benefits. Some of these aforementioned treatments could be good options for preventing NAFLD development. They are easy to use and widely available; thus, it is important that patients and doctors are aware of their existence, an awareness which this study intended to promote. On the other hand, there are new, specialized methods which require research on larger numbers of participants over a longer test time. This paper may help researchers to select one of these methods so as to conduct studies on a larger scale. Our proposed strategies for the treatment of NAFLD are shown in Figure 1.

## Figures and Tables

**Figure 1 jcm-12-01852-f001:**
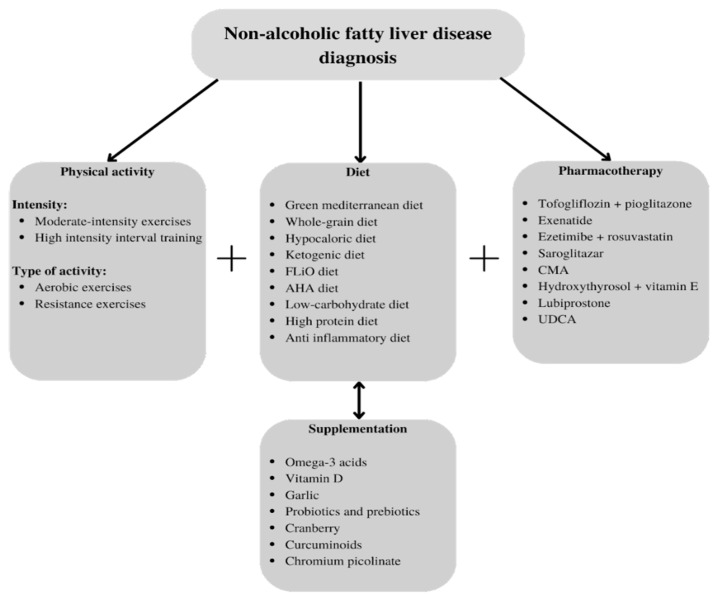
Proposed therapeutic strategies for NAFLD. AHA—American Heart Association; CMA—combined metabolic activator; FLiO—Fatty Liver in Obesity; UDCA—ursodeoxycholic acid.

**Table 1 jcm-12-01852-t001:** The efficiency of natural compound supplementation in NAFLD patients.

Author	Diet	Duration	Results
Tutunchi et al. [50]	oleoyl ethanol amide	12 weeks, 125 mg twice a day	weight ↓, TG ↓, LDL ↓, ALT ↓, AST ↓, BMI ↓, PPAR-α ↑, UCP1 ↑, UCP2 ↑
Medina-Urrutia et al. [51]	chia	8 weeks of 25 g/day	BMI ↓, girth of waist ↓, visceral abdominal fat ↓
Sangouni et al. [52]	garlic powder	12 weeks, 4 tablets, with 400 mg/day	liver fat ↓, improvement of hepatic enzymes and lipid profile
Sangouni et al. [53]	garlic powder	3 months, 1600 mg/day	improvement in insulin resistance and fatty liver index
Kruse et al. [54]	rapeseed oil in combination with an isocaloric diet	8 weeks, 50 g/day	liver fat ↓, free fatty acids ↓, IL-6 ↑
Atefi et al. [55]	sesame oil with hypocaloric diet	12 weeks, 30 g/day	liver fat ↓, liver enzymes ↓, anthropometric parameters ↓
Masnadi Shirazi et al. [56]	cranberry extract	6 months, 144 mg/day	ALP ↓, total cholesterol ↓, TG ↓, improvements in insulin resistance, liver steatosis ↓
Hormoznejad et al. [57]	cranberry extract with weight loss diet	12 weeks, 144 mg twice a day	ALT ↓, improvement in insulin resistance
Goodarzi et al. [58]	pomegranate extract	12 weeks, 225 mg of dried pomegranate twice a day	total cholesterol ↓, TG ↓, LDL/HDL ratio ↓, fasting blood glucose ↓, improvement in insulin resistance, diastolic blood pressure ↓, weight ↓, BMI ↓, waist circumference ↓, HDL ↑
Namkhah et al. [59]	naringenin	4 weeks, 200 mg/day	weight ↓, BMI ↓, improvement in the lipid profile, improvement in NAFLD state
Yari et al. [60]	hesperidin flaxseed	12 weeks, 500 mg twice a day 12 weeks, 30 g/day	TG ↓, LDL ↓, liver steatosis ↓, TNF-α ↓, hs-CRP ↓ HDL ↑, NF-kB ↓, alleviation of liver fibrosis
Saberi-Karimian et al. [61]	curcumine–piperine complex	8 weeks 500 mg/day	TNF-α ↓, MCP-1 ↓, EGF ↓
Mirhafez et al. [62]	curcumin	2 months, 250 mg/day	liver fat ↓, improvements in liver enzymes and lipid profile
Cicero et al. [63]	curcumin	8 weeks, 800 mg twice a day	improvement in fasting plasma insulin, HOMA-IR ↓, waist circumference ↓, blood pressure ↓, TG ↓, HDL↑, liver transaminases ↓, liver steatosis ↓, cortisol ↓
Moradi Kelardeh et al. [64]	resistance training with curcumin	12 weeks, 80 mg/day	ALT ↓, AST ↓
Tutunchi et al. [65]	oleoyl ethanol amide	12 weeks, 125 mg twice a day	NF-κB ↓, IL-6 ↓, IL-10 ↑, fat ↓
Amerikanou et al. [66]	mastiha	6 months, 0.35 g three times a day	improvement in lipid levels, improvement in liver fibrosis
Amerikanou et al. [67]	mastiha	6 months, 2.8 g/day	increase in miR-155 was prevented by mastiha,
Kanoni et al. [68]	mastiha	6 months, 2.1 g/day	total antioxidant status ↑
Izadi et al. [69]	sour tee	8 weeks, 450 mg/day	ALT ↓, AST ↓, TG ↓, diastolic blood pressure ↓, improvement in antioxidant factors
Hosseinabadi et al. [70]	green coffee extract	8 weeks, 400 mg/day	HDL-levels ↑, BMI ↓, weight ↓, improvement in lipid profile, weight ↓
Pervez et al. [71]	delta-tocotrienol	24 weeks, 300 mg twice a day	hepatic steatosis ↓, HOMA-IR ↓, hs-CRP ↓, ALT ↓, AST ↓
Pervez et al. [1]	delta-tocotrienol	48 weeks, 300 mg twice a day	hepatic steatosis ↓, HOMA-IR ↓, serum malondialdehyde ↓
Majnooni et al. [72]	artichoke leaf extract in co-administration with metformin and vitamin E	12 weeks, 400 mg twice a day	ALT ↓, AST ↓, improvement in fatty liver grades
Soleimani et al. [73]	propolis	4 months, 250 mg twice a day	liver fat ↓, fibrosis ↓
Kazemi et al. [74]	sumac	12 weeks, 2000 mg/day	HOMA-IR ↓, improvement in insulin sensitivity index, fasting glucose level ↓, HbA1c ↓, hs-CRP ↓
Jinato et al. [75]	litchi extract	24 weeks, 200 mg/day	improvement in liver steatosis, alteration in intestinal microbiota

ALT—alanine aminotransferase; AST—aspartate aminotransferase; BMI—body mass index; EGF—epidermal growth factor; HbA1c—glycated hemoglobin; HDL—high-density lipoprotein; HOMA-IR—homeostatic model assessment of insulin resistance; hs-CRP—high-sensitivity C-reactive protein; IL—interleukin; LDL—low-density lipoprotein; MCP-1—monocyte chemoattractant protein-1; NF-kB—nuclear factor kappa-light-chain-enhancer of activated B cells; PPAR-α—peroxisome proliferator-activated receptor (PPAR)-alpha; TG—triglycerides; TNF-α—tumor necrosis factor-alpha; UCP—uncoupling protein; ↑—increase; ↓—decrease.

**Table 2 jcm-12-01852-t002:** The efficiency of the chosen dietary approaches for NAFLD.

Author	Type of Study	Intervention Group	Diet	Duration	Results
Montemayor et al. [26]	Multi-center prospective randomized trial	57 patients diagnosed with NAFLD using MRI and MetS, according to the International Diabetes Federation (IDF)	Mediterranean diet (high adherence) in connection with physical activity promotion and 25–30% reduction in baseline calorie intake	6 months	↑: HDL-c (NS) ↓: BMI (S), BW (S), WC (S), SBP (S), DBP (S), HFC (S), glucose level (NS), TG (S), HOMA-IR (NS), AST (NS), ALT (NS), GGT (S)
Cohen et al. [82]	RCT	16 adolescent patients with an NAFLD diagnosis proven by biopsy	Low-free-sugar diet	8 weeks	↓: Hepatic DNL (S), HFC (NS), fasting insulin (NS), ALT (S)
Dorosti et al. [34]	RCT	47 patients with NAFLD diagnosed by ultrasonography and liver fat content	Whole-grain diet	12 weeks	↑: HDL (S) ↓: Ultrasound fatty liver grade (S), ALT (S), AST (S), GGT (S), TC (S), LDL (S), TG (NS), SBP (NS), DBP (NS), serum insulin (S), HOMA-IR (S)
Moradi et al. [81]	RCT	23 patients with NAFLD diagnosed by ultrasound	Chromium picolinate, 400 μg/day supplementation	3 months	↑: QUICKI (S) ↓: TG (S), AIP (S), VLDL-c (S), HOMA-IR (S), hs-CRP (S), IL-6 (S), TNF-α (S), fetuin-A (S)
Medina-Urrutia et al. [51]	Single-arm experimental design study	25 patients with NAFLD diagnosed by computer tomography	Milled Salvia hispanica (chia) at a dose of 25 g/day, supplemented in an isocaloric diet	8 weeks	↓: TC (S), non-HDL-c (S), FFA (S), BW (S), BMI (S), WC (S), SAT (S), VAT (S)
Hormoznejad et al. [57]	RCT	25 patients with NAFLD diagnosed by ultrasound	Weight loss diet plus cranberry supplementation, 2 tablets each containing 144 mg of Vaccinium macrocarpon extract	12 weeks	↑: Muscle mass (S) ↓: BW (S), body fat (S), BMI (S), WC (S), ALT (S), AST (NS), GGT (NS), insulin (S), HOMA-IR (S), TG (NS), TC (NS), LDL-c (NS)
Sangouni et al. [52]	RCT	45 patients with fatty liver grade 1 to 3 based on ultrasound	Garlic powder supplementation—400 mg 4 times a day	12 weeks	↓: Hepatic steatosis (S), ALT (S), AST (S), GGT (S), TG (S), HDL (S), LDL (S)
Soleimani et al. [73]	RCT	27 patients with hepatic steatosis diagnosed by elastography	Propolis, 250 mg twice/day	4 months	↓: Liver stiffness (S), hs-CRP (S), BW (NS), TG (NS), TC (NS)

Adipo-IR—measure of adipose tissue insulin resistance; AIP—atherogenic index of plasma; ALT—alanine aminotransferase; AST—aspartate aminotransferase; BW—body weight; BMI—body mass index; CAP—hepatic-steatosis-index-controlled attenuation parameter; CIMT—carotid intima-media thickness; DNL—de novo lipogenesis; FCP—fasting C peptide; FLI—fatty liver index; FFA—free fatty acid; FIB-4—fibrosis-4 index; GGT—γ-glutamyl transpeptidase; HDL—high-density lipoprotein; HFC—hepatic fat content; HFF—hepatic fat fraction; HOMA-IR—homeostatic model assessment of insulin resistance; hs-CRP—high-sensitivity C-reactive protein; IL-6—interleukin 6; LDL—low-density lipoprotein; METs—metabolic syndrome; QUICKI—quantitative insulin sensitivity check index; RCT—randomized controlled trial; rhGH—recombinant human growth hormone; RLP-c—remnant-like particle cholesterol; SAT subcutaneous adipose; SBP—systolic blood pressure; TC—total cholesterol; TNF-α—tumor necrosis factor-alpha; VAT—visceral adipose tissue; VLDL-C—very-low-density lipoprotein cholesterol; WC—waist circumference. ↑—increase; ↓—decrease. S—statistically significant. NS—statistically non-significant.

**Table 3 jcm-12-01852-t003:** The efficiency of chosen treatment options for NAFLD.

Study	Type of Study	Intervention Group	Medication	Duration	Results	Safety
Phrueksotsai et al. [106]	RCT	18 patients with DM2 and hepatic steatosis confirmed by abdominal ultrasonography or CT	Dapagliflozin, 10 mg per day	12 weeks	↑: HDL (NS), adiponectin (S) ↓: HFC (S), weight (S), BMI (S), body fat (S), ALT (S), HbA1c (S), HOMA-IR (NS), uric acid (NS), LDL (NS)	Lack of significant adverse effects
Yoneda et al. [104]	Randomized, prospective, open-label controlled trial	21 patients with DM2 and NAFLD measured by hepatic fat fraction of at least 10%, as assessed based on the MRI-proton density fat fraction	Tofogliflozin, 20 mg per day	24 weeks	↑: HDL (S), ketone bodies (S) ↓: HFC (S), BW (S), ALT (S), AST (S), GGT (S), liver stiffness—MRE-LSM (NS), HbA1c (S), uric acid (S), oxidative stress (S), hepatocyte apoptosis—CK-18 fragment M30 antigen (S)	One case of urinary tract infection, lack of life-threatening events
Taheri et al. [107]	RCT	43 patients with an NAFLD diagnosis based on previous ultrasound imaging or liver function test without diabetes comorbidity	Empagliflozin, 10 mg/day	24 weeks	↓: liver stiffness (S), liver steatosis (S), AST (S), ALT (S), fasting insulin (S), BW (S), BMI (S), WC (S)	Lack of major adverse effects
Takahashi et al. [105]	RCT	25 patients with DM and NAFLD diagnosed by liver biopsy	Ipragliflozin, 50 mg/day	72 weeks	↓: HbA1c (S), BMI (S), fasting glucose (S), VAT(S), SAT (S), AST (S), ALT (S), GGT (S), type IV collagen 7s—marker of fibrosis (S), liver fibrosis reduction by at least one stage (S)	Mild to moderate adverse effects reported by 22.2% of participants
Yoneda et al. [104]	Randomized, prospective, open-label controlled trial	19 patients with DM2 and NAFLD measured by hepatic fat fraction of at least 10%, as assessed based on the MRI-proton density fat fraction	Pioglitazone, 15–30 mg per day	24 weeks	↑: BW (S), HDL (S), adiponectin (S) ↓: HFC (S), ALT (S), AST (S), GGT (S), alkaline phosphatase levels (S), liver stiffness—MRE-LSM (S), HbA1c (S), TG (S), oxidative stress (NS), hepatocyte apoptosis—CK-18 fragment M30 antigen (S)	Adverse effects: edema and weight gain, lack of life-threatening events
Wang et al. [113]	A prospective comparative study	14 patients with DM2 comorbidity with ab NAFLD diagnosis based on fulfillment of diagnostic criteria	Sitagliptin, 100 mg/day	24 weeks	↓: HFC (NS), HbA1c (S), FPG (S), WC (S), BMI (S)	Well tolerated
Liu et al. [115]	RCT	38 patients with NAFLD measured by proton MRS and newly diagnosed DM2	Exenatide s.c. 5 μg twice daily for 4 weeks followed by 10 μg twice daily for 20 weeks	24 weeks	↓: HFC (S), FIB-4 (S), VAT (S), SAT (S), ALT, AST, GGT, TC (S), TG (S) LDL-c (S), FFA (S), WC (S), BW (S), BMI (S), SBP (S), DBP (S)	Hypoglycemic events, lack of major adverse effects
Flint et al. [116]	RCT	34 patients with NAFLD measured by MRI-PDFF	Semaglutide s.c. 0.4 mg/day	72 weeks	↓: liver steatosis (S), hepatic fat volume (S), total liver volume (S), VAT (S), SAT (S), BW (S), HbA1c (S), ALT (S), AST (S), GGT (S), SBP (S), hs-CRP (S), TG (S)	Reported adverse reactions were similar in the treatment group and placebo; gastrointestinal effects and serious adverse effects reported by 12.1% of subjects
Guo et al. [120]	RCT	Patients with DM2 and NAFLD indicated by hepatic steatosis upon imaging or histology	Liraglutide s.c. at onset 0.6 mg/day, increased weekly with forced titration to 1.8 mg plus metformin at 2 g/day	26 weeks	↓: HFC (S), SAT (S), VAT (S), AST (S), ALT (S), HOMA-IR (S), BW (S), WC (S), BMI (S), HbA1c (NS), FBG (NS)	Events of mild hypoglycemia, nausea, diarrhea and vomiting
Liu et al. [115]	RCT	38 patients with NAFLD measured by proton MRS and newly diagnosed DM2	Insulin glargine (Lantus) at doses needed to achieve below 7.0 mmol/L of FPG	24 weeks	↓: HFC (S), VAT (NS), ALT (S), GGT (S), HbA1c (S), TG (S), FFA (S), WC (S), FCP (S)	Hypoglycemic events, lack of major adverse effects
Nakajima et al. [124]	RCT	58 patients with MRI-PDFF proven elevated liver fat content	Pemafibrate, 0.2 mg twice a day	72 weeks	↓: MRE-based liver stiffness (S), HFC (NS), ALT (S), GGT (S), ALP (S), LDL-c (S), TC (S), TG (S), HDL-c (S)	Therapy well tolerated, mild and moderate severity adverse effects
Cho et al. [123]	RCT	34 patients with liver steatosis proven by ultrasound with or without DM2	Ezetimibe 10 mg/day plus rosuvastatin 5 mg/day	24 weeks	↓: HFC (S), CAP (S), BMI (S), WC (S), LDL-c (S), TG (S), CRP (S)	Lack of significant adverse effects
Nadinskaia et al. [126]	Open-label, multicenter, international noncomparative trial	74 patients with NAFLD diagnosed by ultrasound	Ursodeoxycholic acid 15 mg/kg/day	6 months	↑: HDL (S)—only in women’s group ↓: BW (S), FLI (S), TC (S), LDL (S), TG (S), CIMT (S), 10-year ASCVD risk (S—for women, NS—for men), ALT (S for man and women), ALT (S for man), AST (S for men), GGT (S for men)	No data
Pan et al. [128]	Randomized open label trial	young people with ≥5% hepatic fat fraction on proton magnetic resonance spectroscopy	Somatropin at a starting dose of 0.5 mg/day with further titration to IGF-1 z-score	24 weeks	↓: HFF (S), BMI (S), ALT (NS), AST (NS), GGT (NS)	Lack of treatment-related reasons to discontinue the study
Climax et al. [141]	RCT	33 patients with NAFLD diagnosed by imaging or histology	Epeleuton at a dose of 1 g twice a day	16 weeks	↓: HFC (NS), ALT (NS), HOMA-IR (S), Adipo-IR (S), HbA1c (S), FPG (S), TG (S), TC (S), VLDL-c (S), RLP-c (S), non–HDL-C (NS), circulating inflammatory markers (S)	45.5% of responders reported adverse events connected with treatment being mild to moderate in severity but probably not related to epeleuton, lack of serious adverse effects
Zeybel et al. [99]	Randomized, placebo-controlled phase 2 study	20 overweight patients with NAFLD diagnosis—liver fat was determined by MRI-PDFF	Combined metabolic activators (CMAs) containing 3.73 g L-carnitine tartrate, 1 g nicotinamide riboside, 12.35 g serine and 2.55 g N-acetyl-l-cysteine)—one dose for first 14 days, followed by 2 doses up to the end of the study	10 weeks	↓: HFC (S), ALT (S), AST (S), uric acid (S), creatinine (S), SBP (S), inflammatory protein markers (S)	12 of 20 subjects reported mild–moderate adverse symptoms, but they decided to be followed up in the study
Harrison et al. [102]	Randomized clinical study	29 patients with NAFLD diagnosis indicated by CT, MR and AST level, including 12 patients with diabetes comorbidity	AXA1125 24 g twice daily	16 weeks	↓: HFC (S), HOMA-IR (NS), ALT (NS), liver fibrosis (NS), apoptosis marker—K-18 M65 (S—after 8 weeks, NS—at the endpoint), cT1—imaging marker of inflammation and fibrosis (S)	Mild or moderate adverse effects, 1 subject resigned from the study

Adipo-IR—measure of adipose tissue insulin resistance; ALT—alanine aminotransferase; AST—aspartate aminotransferase; BMI—body mass index; BW—body weight; CAP—hepatic-steatosis-index-controlled attenuation parameter; CIMT—carotid intima-media thickness; FCP—fasting C peptide; FLI—fatty liver index; FFA—free fatty acid; FIB-4—fibrosis-4 index; FPG—fasting plasma glucose; GGT—γ-glutamyl transpeptidase; HDL—high-density lipoprotein; HFC—hepatic fat content; HFF—hepatic fat fraction; HOMA-IR—homeostatic model assessment of insulin resistance; hs-CRP—high-sensitivity C-reactive protein; LDL—low-density lipoprotein; RCT—randomized controlled trial; rhGH—recombinant human growth hormone; RLP-c—remnant-like particle cholesterol; SAT subcutaneous adipose; TC—total cholesterol; VAT—visceral adipose tissue; VLDL-C—very-low-density lipoprotein cholesterol; WC—waist circumference. ↑—increase, ↓—decrease. S—statistically significant. NS—statistically non-significant.

## Data Availability

Not applicable.

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
