# Peer review of "What’s New in the Treatment of Non-Alcoholic Fatty Liver Disease (NAFLD)"

_jcm, 2023, doi:10.3390/jcm12051852_

Round 1
Reviewer 1 Report
Authors described in detail pharmacological options of the treatment in the course of NAFLD. Firstly, I would suggest to use the term of MAFLD instead of NAFLD - according to the newest nomenclature. Additionally, what I do miss in this paper is mentioning the role of miRNAs in the course of MAFLD and presenting them as a possible target for the treatment. All in all it highlights the most important up-to-date pathways in the management of MAFLD.
Author Response
Response to Reviewer 1 Comments
Thank the Reviewer’s very much for your time and valuable comments on our manuscript. The responses for all points are below. The changes were introduced into the text of manuscript, as suggested by the Reviewer.
Point 1: Authors described in detail pharmacological options of the treatment in the course of NAFLD. Firstly, I would suggest to use the term of MAFLD instead of NAFLD - according to the newest nomenclature.
Response 1: Thank the Reviewer’s suggestion. According to it, we made the changes in the manuscript in place, where we can do it.
Point 2: Additionally, what I do miss in this paper is mentioning the role of miRNAs in the course of MAFLD and presenting them as a possible target for the treatment.
Response 2: Thank the Reviewer’s suggestion. We added such a description in lines 1010-1019.
In addition, we modified our article according to the suggestions of other reviewers.
We sincerely hope that all changes introduced by us in the text will be fully satisfactory for the Reviewer.

Reviewer 2 Report
This is a very comprehensive review article that due to its comprehensiveness is very difficult to read and follow. My main impression is that it has to be shortened for at least 40% and some of the data needs to be presented in tables for the sake of easier understanding of the main points;
Few other suggestions
1. Line 39- please add diverticulosis ( Pantic I, Lugonja S, Rajovic N, Dumic I, Milovanovic T. Colonic Diverticulosis and Non-Alcoholic Fatty Liver Disease: Is There a Connection? Medicina (Kaunas). 2021 Dec 27;58(1):38. doi: 10.3390/medicina58010038. PMID: 35056346; PMCID: PMC8778461.)
2. Line 43-44: While I think it is ok to keep term NAFLD please do add that the term change has been proposed to MAFLD to illustrate metabolic component more clearly ( Yamamura S, Eslam M, Kawaguchi T, Tsutsumi T, Nakano D, Yoshinaga S, Takahashi H, Anzai K, George J, Torimura T. MAFLD identifies patients with significant hepatic fibrosis better than NAFLD. Liver Int. 2020 Dec;40(12):3018-3030. doi: 10.1111/liv.14675. PMID: 32997882.)
3. Line 59-65- please delete, it deviates from the topic and introduction is already too long
4. Line 69 to 76 also should be reduced
5. Methodology- please report PRISMA diagram that illustrates article selection criteria and why initially selected article were excluded etc
6. Diet and supplement section has to be reduced for at least 50%; It is very redundant and too long.
7. Role of vitamin E has not been explored despite it being effective in females with NAFLD : " Perumpail BJ, Li AA, John N, Sallam S, Shah ND, Kwong W, Cholankeril G, Kim D, Ahmed A. The Role of Vitamin E in the Treatment of NAFLD. Diseases. 2018 Sep 24;6(4):86. doi: 10.3390/diseases6040086. PMID: 30249972; PMCID: PMC6313719."
8. Pharmacotherapy- authors should mention interrelation between metabolic conditions and that improvement in one of the related conditions might lead to improvement in others- please see Milovanovic T, Pantic I, Dragasevic S, Lugonja S, Dumic I, Rajilic-Stojanovic M. The Interrelationship Among Non-Alcoholic Fatty Liver Disease, Colonic Diverticulosis and Metabolic Syndrome. J Gastrointestin Liver Dis. 2021 Jun 18;30(2):274-282. doi: 10.15403/jgld-3308. PMID: 33951119.
Author Response
Response to Reviewer 2 Comments
Thank the Reviewer’s very much for your time and valuable comments on our manuscript. The responses for all points are below. The changes were introduced into the text of manuscript, as suggested by the Reviewer.
This is a very comprehensive review article that due to its comprehensiveness is very difficult to read and follow. My main impression is that it has to be shortened for at least 40% and some of the data needs to be presented in tables for the sake of easier understanding of the main points;
Thank the Reviewer’s suggestion. As recommended, we significantly reduced the text and introduced Table 1.
Point 1: Line 39- please add diverticulosis ( Pantic I, Lugonja S, Rajovic N, Dumic I, Milovanovic T. Colonic Diverticulosis and Non-Alcoholic Fatty Liver Disease: Is There a Connection? Medicina (Kaunas). 2021 Dec 27;58(1):38. doi: 10.3390/medicina58010038. PMID: 35056346; PMCID: PMC8778461.)
Response 1: Thank the Reviewer’s suggestion. We have added reference number 4.
Point 2: Line 43-44: While I think it is ok to keep term NAFLD please do add that the term change has been proposed to MAFLD to illustrate metabolic component more clearly ( Yamamura S, Eslam M, Kawaguchi T, Tsutsumi T, Nakano D, Yoshinaga S, Takahashi H, Anzai K, George J, Torimura T. MAFLD identifies patients with significant hepatic fibrosis better than NAFLD. Liver Int. 2020 Dec;40(12):3018-3030. doi: 10.1111/liv.14675. PMID: 32997882.)
Response 2: Thank the Reviewer’s suggestion. As recommended, changes were made to the manuscript (lines 45-53) and reference number 7 was added
Point 3: Line 59-65- please delete, it deviates from the topic and introduction is already too long.
Response 3: Thank the Reviewer’s suggestion. Changes made as recommended in the manuscript.
Point 4: Line 69 to 76 also should be reduced
Response 4: Thank the Reviewer’s suggestion. Changes made as recommended in the manuscript.
Point 5: Methodology- please report PRISMA diagram that illustrates article selection criteria and why initially selected article were excluded etc.
Response 5: Thank the Reviewer’s suggestion. According to the recommendations, the PRISMA diagram was added and placed in the supplementary materials (suplementary table 1 and suplementary figure 1) and the description in the manuscript in the Materials and Methods chapter (lines from 107).
Point 6: Diet and supplement section has to be reduced for at least 50%; It is very redundant and too long.
Response 6: Thank the Reviewer’s suggestion. These subsections have been shortened as recommended and Table 1 has been added.
Point 7: Role of vitamin E has not been explored despite it being effective in females with NAFLD : " Perumpail BJ, Li AA, John N, Sallam S, Shah ND, Kwong W, Cholankeril G, Kim D, Ahmed A. The Role of Vitamin E in the Treatment of NAFLD. Diseases. 2018 Sep 24;6(4):86. doi: 10.3390/diseases6040086. PMID: 30249972; PMCID: PMC6313719."
Response 7: Thank the Reviewer’s suggestion. as suggested, we added changes to the text (line 739-742). Unfortunately, the proposed references were not added as they did not meet the inclusion criteria under PRISMA. However, we adapted to this issue based on other references - number 97.98.
Point 8: Pharmacotherapy- authors should mention interrelation between metabolic conditions and that improvement in one of the related conditions might lead to improvement in others- please see Milovanovic T, Pantic I, Dragasevic S, Lugonja S, Dumic I, Rajilic-Stojanovic M. The Interrelationship Among Non-Alcoholic Fatty Liver Disease, Colonic Diverticulosis and Metabolic Syndrome. J Gastrointestin Liver Dis. 2021 Jun 18;30(2):274-282. doi: 10.15403/jgld-3308. PMID: 33951119.
Response 8: Thank the Reviewer’s suggestion. As noted, we added a paragraph (lines 726-731) and reference number 93.
We sincerely hope that all changes introduced by us in the text will be fully satisfactory for the Reviewer.

Round 2
Reviewer 2 Report
I would like to thank the authors for detailed revisions. In my opinion the paper has been improved, and I do not have any further comments.